# Flavin-enabled reductive and oxidative epoxide ring opening reactions

Bidhan Chandra De[1,2,9], Wenjun Zhang[1,2,3,4,9], Chunfang Yang[1,2,3,4], Attila Mándi[5], Chunshuai Huang[1], Liping Zhang[1,2,3], Wei Liu[1], Mark W. Ruszczycky[6], Yiguang Zhu[1,2,3,4], Ming Ma [7], Ghader Bashiri [8], Tibor Kurtán[5], Hung-wen Liu [6] ✉ & Changsheng Zhang [1,2,3,4] ✉

Epoxide ring opening reactions are common and important in both biological processes and synthetic applications and can be catalyzed in a non-redox manner by epoxide hydrolases or reductively by oxidoreductases. Here we report that fluostatins (FSTs), a family of atypical angucyclines with a benzo-fluorene core, can undergo nonenzyme-catalyzed epoxide ring opening reactions in the presence of flavin adenine dinucleotide (FAD) and nicotina-mide adenine dinucleotide (NADH). The 2,3-epoxide ring in FST C is shown to open reductively via a putative enol intermediate, or oxidatively via a per-oxylated intermediate with molecular oxygen as the oxidant. These reactions lead to multiple products with different redox states that possess a single hydroxyl group at C-2, a 2,3-vicinal diol, a contracted five-membered A-ring, or an expanded seven-membered A-ring. Similar reactions also take place in both natural products and other organic compounds harboring an epoxide adjacent to a carbonyl group that is conjugated to an aromatic moiety. Our findings extend the repertoire of known flavin chemistry that may provide new and useful tools for organic synthesis.

Epoxides are important building blocks in organic synthesis and biosynthesis[1,2]. Due to the significant strain of the three-membered oxirane ring and polarized oxygen-carbon bonds, epoxides can readily undergo regioselective and stereoselective ring-opening under the control of an appropriate catalyst[3,4]. Epoxide ring opening generally proceeds via nucleophilic addition with inversion of stereochemistry; however, the regioselectivity can be sensitive to whether the reaction is conducted under acidic or alkaline conditions as shown in Fig. 1a[5]. Accordingly, a number of nucleophiles have been developed to react with epoxides in order to yield 1,2-difunctionalized systems with *trans*

stereochemistry[6]. Epoxide ring-opening can also be catalyzed by enzymes such as epoxide hydrolases (EHs) or oxidoreductases (ORs) (Fig. 1b)[7]. EHs catalyze the hydrolysis of epoxides to *trans*-vicinal diols[8,9]; ORs catalyze the stereoselective reduction of an epoxide to an alcohol (Fig. 1b)[10,11].

Epoxides are also an important structural element found in a wide variety of natural products and reactive intermediates in biosynthetic pathways[1]. In particular, the fluostatins (FSTs, e.g. **1**)[12–22], kinamycins[23], lomaiviticins[24,25], and nenestatins[26–28] constitute a family of atypical angucyclines with a benzofluorene core (Supplementary Fig. 1). The

[1]Key Laboratory of Tropical Marine Bioresources and Ecology, Guangdong Key Laboratory of Marine Materia Medica, South China Sea Institute of Oceanology, Chinese Academy of Sciences, Guangzhou 510301, China. [2]University of Chinese Academy of Sciences, 19 Yuquan Road, Beijing 100049, China. [3]Southern Marine Science and Engineering Guangdong Laboratory (Guangzhou), 1119 Haibin Road, Nansha District, Guangzhou 511458, China. [4]Sanya Institute of Ocean Eco-Environmental Engineering, Yazhou Scientific Bay, Sanya 572000, China. [5]Department of Organic Chemistry, University of Debrecen, P.O. Box 400, H-4002 Debrecen, Hungary. [6]Division of Chemical Biology & Medicinal Chemistry, College of Pharmacy, and Department of Chemistry, University of Texas at Austin, Austin, TX 78712, USA. [7]State Key Laboratory of Natural and Biomimetic Drugs, School of Pharmaceutical Sciences, Peking University, 38 Xueyuan Road, Haidian District, Beijing 100191, China. [8]Laboratory of Molecular and Microbial Biochemistry, School of Biological Sciences, The University of Auckland, Auckland, New Zealand. [9]These authors contributed equally: Bidhan Chandra De, Wenjun Zhang. ✉e-mail: h.w.liu@mail.utexas.edu; czhang@scsio.ac.cn

**Fig. 1 | Representative epoxide ring-opening reactions. a** Nonenzymatic epoxide ring-opening reactions involving addition of a nucleophile (Nuc) under basic or acidic conditions. **b** Enzymatic epoxide ring-opening reactions by epoxide hydrolases (EHs) and oxidoreductases (ORs). **c** Epoxide ring-opening reactions of fluostatin C (**1**), either enzymatically catalyzed by the hydrolase Alp1U, or none-nzymatically mediated by FAD and NADH as reported in this study.

benzofluorene core of the kinamycins and lomaiviticins is further decorated with a diazo group endowing these compounds with potent antitumor activities that has attracted considerable attention[29,30]. The atypical angucyclines are often characterized by a highly oxygenated A-ring that can undergo further modifications such as acylation, epoxidation, glycosylation, and dimerization leading to significant structural diversity[18,21,27,31,32]. The α/β hydroxylases Alp1U and Lom6 were recently reported to catalyze stereoselective epoxide hydrolysis reactions during the biosynthesis of kinamycins and lomaiviticin[23]. Alp1U has also been shown to hydrolyze the epoxide of FST C (**1**) to afford the two stereoisomeric FSTs C1 (**2**) and C2 (**3**) (Fig. 1c)[18,33].

The α/β-hydrolase FlsH from the FST pathway in the South China Sea-derived *Micromonospora rosaria* SCSIO N160 has been shown to catalyze deacylation of acyl FSTs;[18] however, it is unable to catalyze hydrolysis of the epoxide in FSTs despite its homology to Alp1U[18]. Naturally isolated FSTs (Supplementary Fig. 1), such as FST B[12], pyr-azolofluostatins A–C[16], and FST R[17] have been proposed to derive from the corresponding epoxide precursors; however, given the activity of FlsH it is not entirely clear how the epoxide would be cleaved and what enzymes if any would be responsible.

In this work, the 2,3-epoxide ring in FST C (**1**) is shown to open nonenzymatically in the presence of flavin adenine dinucleotide (FAD, **4**) and nicotinamide adenine dinucleotide (NADH) with reduced FADH₂ (**5**) or C4a-hydroperoxyflavin (FADOOH, **6**) serving as the reactive flavin species (Fig. 1c). The epoxide ring-opening reactions can thus proceed in a reductive manner to generate a putative enol intermediate, which is tautomerized to products possessing a single hydroxyl group at C-2 (**7** and **8**), or is further modified by **6** to products with a vicinal diol at C-2 and C-3 (**3** and **9**). In the presence of molecular oxygen, the epoxide ring opening reactions can also proceed in an oxidative manner leading to a peroxylated

intermediate, which can be further reduced by NADH to **3** and **9**, or can undergo ring rearrangements to afford products with a con-tracted five-membered A-ring (**10**) or a bridged and expanded seven-membered A-ring (**11**). As a result, this non-enzymatic epoxide clea-vage reaction may account for the formation of several different FST derivatives observed in Nature. Additional experiments show that similar reactions also take place in both natural products and other organic compounds harboring an epoxide adjacent to a carbonyl group that is conjugated to an aromatic moiety. Thus, this none-nzyme catalyzed ring-opening reaction holds promise as a useful tool in organic synthesis.

## Results

### Nonenzymatic epoxide ring-opening reaction of FST C (1)

The α/β hydroxylase Alp1U was previously shown to hydrolyze the epoxide of FST C (**1**) to afford FST C1 (**2**) and FST C2 (**3**) (Fig. 2a, traces i-ii)[18,33]. In contrast, the Alp1U homologue FlsH is unable to catalyze hydrolysis of the epoxide ring in **1** (Fig. 2a, trace iii)[18]. A possible explanation is that FlsH might require a protein partner or an exogenous cofactor to catalyze the epoxide hydrolysis. In order to test these hypotheses, several flavin reductases including Fre from *E. coli*[34] were selected to examine their competence as putative protein partners for FlsH. It was found that incubation of FST C (**1**) with FlsH and Fre in the presence of NADH led to multiple products (Fig. 2a, trace vi). However, further experiments showed that formation of the same set of products was also observed in the absence of FlsH (Fig. 2a, trace vii), suggesting that Fre alone was able to mediate the same decomposition of FST C (**1**). Moreover, incubation of **1** with several other flavoenzymes including FlsO2 (prejadomycin oxidase)[35], TiaM (tiacumicin halogenase)[36,37], and XiaK (xiamycin *N*-hydroxylase)[38] also led to the same result (Supple-mentary Fig. 2). Given that the activities of these flavoenzymes all rely on

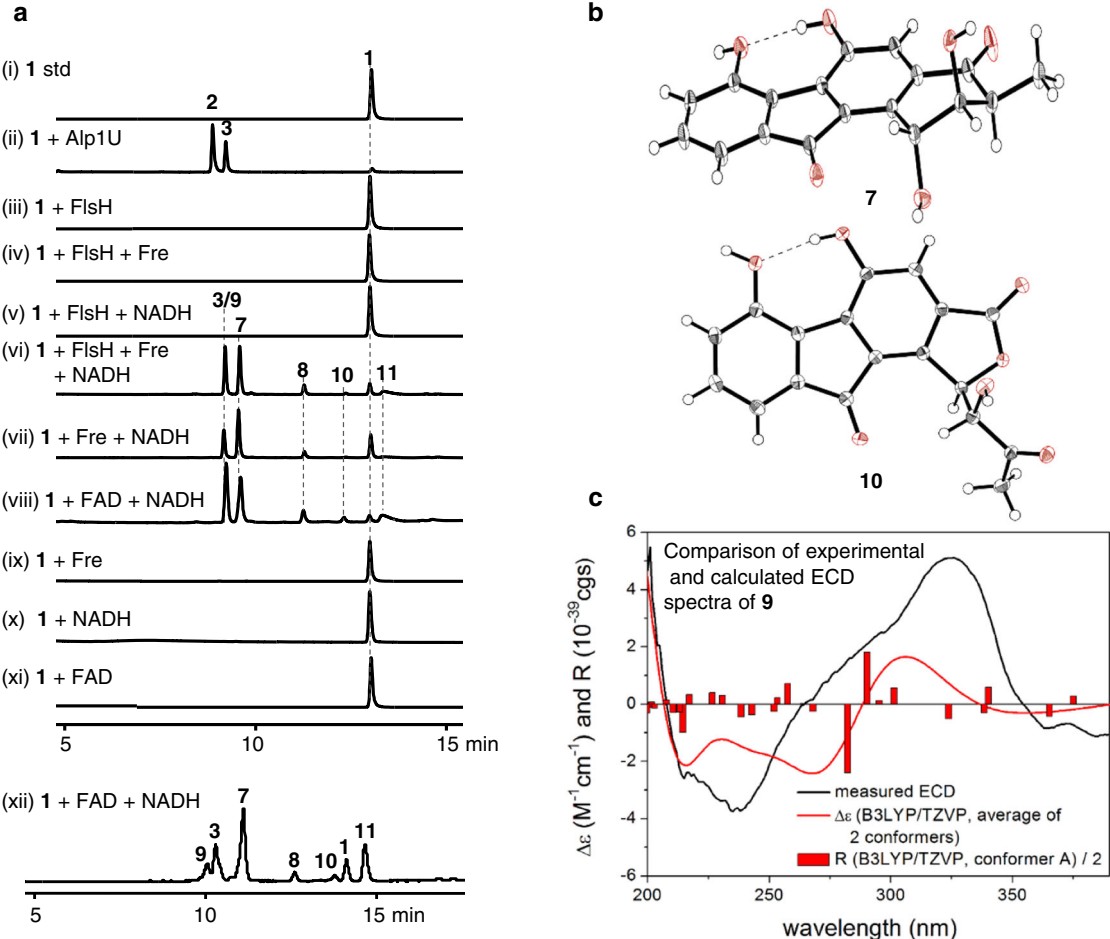

**Fig. 2 | Epoxide ring opening reactions involving FST C (1). a** HPLC analysis of reactions involving FST C (**1**). (i) **1** std.; (ii) **1** + Alp1U; (iii) **1** + FlsH; (iv) **1** + FlsH + Fre; (v) **1** + FlsH + NADH; (vi) **1** + FlsH + Fre + NADH; (vii) **1** + Fre + NADH; (viii) **1** + FAD + NADH; (ix) **1** + Fre; (x) **1** + NADH; (xi) **1** + FAD; (xii) **1** + FAD + NADH. HPLC was performed using a reversed phase C18 column (i–xi) or in a polar column (xii). Reactions were run for 30 min at 30 °C in 50 mM PBS buffer (pH 7.0) containing 5 μM enzyme(s) (Alp1U, FlsH or Fre), 100 μM FAD and 10 mM NADH. **b** The X-ray crystal structures of **7** and **10**. **c** Experimental ECD spectrum of **9** (black line) compared with the B3LYP/TZVP PCM/MeCN // ωB97X/TZVP PCM/MeCN spectrum of (1R,2S,3R)-**9** (red line). The bars represent rotational strength values for the lowest energy solution conformers.

the reduction of FAD by NADH, the reduced flavin alone was proposed to serve as the actual catalyst of FST C (**1**) consumption. Indeed, similar products were observed upon incubation of FST C (**1**) with FAD and NADH in the absence of any enzymes, such that FAD and NADH are both sufficient (Fig. 2a, traces viii & xii) and necessary (Fig. 2a, traces ix–xi) to facilitate the nonenzymatic decomposition of FST C (**1**).

## Identification of the FST C decomposition products

Reaction products (**3** and **7–11**) were isolated from a scaled up reaction of FST C (**1**) with FAD and NADH and structurally characterized as shown in Fig. 1c. The first product was identified to be FST C2 (**3**) (Supplementary Fig. 3 and Table 1), which is also a product of the Alp1U-catalyzed hydrolysis reaction[33]. The second reaction product **7** (Supplementary Fig. 4 and Table 2) was established to have the same planar structure as FST B, an FST naturally isolated from *Streptomyces* sp. TA-3391[12] and *Streptomyces* strain Acta 1383;[13] however, the absolute configuration of FST B was not determined[12]. The (1R,2R,3S) absolute configuration of **7** was established by the solution TDDFT-ECD methodology (Supplementary Fig. 5) and confirmed by single crystal X-ray analysis (Fig. 2b, CCDC 2036399; Supplementary Table 3). The third product **8** (Supplementary Fig. 6 and Table 2) was determined to have the same planar structure as **7** and the (1R,2R,3R) absolute configuration of **8** was assigned by comparison of the experimental and calculated ECD spectra carried out on the (1R,2R,3R)

stereoisomer (Supplementary Fig. 7)[39]. In order to distinguish the two stereoisomers of FST B, **7** and **8** were named FST B1 and FST B2 (Fig. 1), respectively. The product **9** (designated FST C3, Fig. 1) was isolated with the same retention time as that of **3** during in the HPLC analysis with a reversed-phase C18 column. Analysis of the assay using a polar HPLC column allowed separation of these two species (Fig. 2a, trace xii). Finally, **9** was identified to be a stereoisomer of **3** by HRESIMS and NMR data (Supplementary Fig. 8 and Table 4), and the (1R,2S,3S) absolute configuration of **9** was established by TDDFT-ECD methodology (Fig. 2c, Supplementary Fig. 9).

Two additional products **10** and **11** were also identified and structurally characterized (Fig.1c; Fig. 2a, traces vi-viii & xii). The product **10**, designated FST C4 (Fig. 1), was found to possess a contracted five-membered A-ring by NMR spectroscopic analysis (Supplementary Fig. 10 and Table 5) and was determined to have the (1R,2R) absolute configuration by single crystal X-ray analysis (Fig. 2b; Supplementary Table 6; CCDC 2129081). The product **11**, designated FST C5 (Fig. 1), was structurally elucidated to have a bridged and expanded seven-membered A-ring (Supplementary Fig. 11 and Table 5). Assignment of the (1R,2R,3R) absolute configuration to **11** was based on its origin from **1** but otherwise remains tentative, because a definitive experimental characterization was not possible on account of its instability. Currently, there are no reports of **9–11** having been isolated from a natural source.

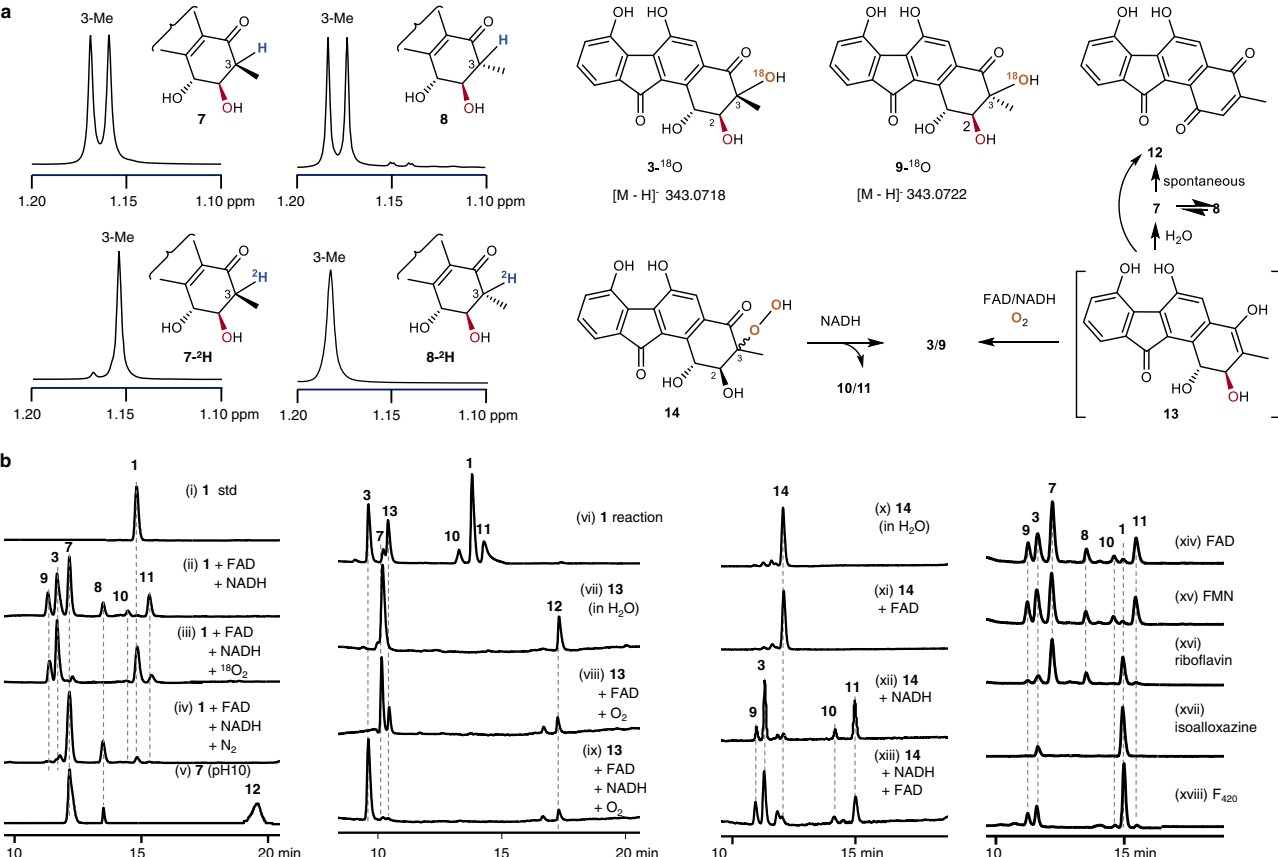

**Fig. 3 | Detailed studies on the epoxide ring opening reactions of 1 by isotope labeling, intermediate characterization and cofactor compatibility. a** NMR analysis of $^2$H or $^{18}$O incorporation in products and intermediates as well as potential interconversions. Deuterium incorporation from $^2$H$_2$O at C-3 of **7** and **8** was inferred from $^1$H NMR spectroscopic analysis. The O$^{18}$-labeling in **3**-$^{18}$O and **9**-$^{18}$O was supported by HRMS analysis. The assigned structures of **12** and **14** are shown, whereas the proposed structure of **13** remains speculative on account of its poor stability. **b** HPLC analysis of reactions under varying conditions. (i) **1** std.; (ii) **1** + FAD + NADH (in air); (iii) **1** + FAD + NADH (under $^{18}$O$_2$); (iv) **1** + FAD + NADH (under N$_2$); (v) **7** in 50 mM borax/NaOH buffer (pH 10) at 30 °C for 12 h; (vi) 100 µM

**1** + 10 µM FAD + 2 mM NADH (30 °C for 30 min); (vii–ix) incubation of **13** collected from (vi) with (vii) H$_2$O; (viii) FAD; (ix) FAD + NADH; the reactions of (viii) and (ix) were performed in O$_2$ at 30 °C for 30 min in 50 mM PBS buffer (pH 7.0); (x) **14** in H$_2$O; (xi) **14** + FAD; (xii) **14** + NADH; (xiii) **14** + FAD + NADH. The reactions of (x-xiii) were performed at 30 °C for 30 min. The reactions of (xiv–xvii) involved incubation of **1** and NADH with different flavin cofactors: (xiv) FAD; (xv) FMN; (xvi) riboflavin; (xvii) isoalloxazine at 30 °C for 30 min. (xviii) The reaction containing **1**, F$_{420}$, glucose 6-phosphate and FGD was incubated at 30 °C for 10 h. HPLC analysis was run with UV detection at 304 nm using a polar column (traces i-v & x-xviii) or a reversed phase C18 column (traces vi-ix).

## Labeling studies of the epoxide ring-opening reaction

To investigate the mechanism of the nonenzymatic epoxide ring opening reaction, FST C (**1**), FAD and NADH were coincubated in PBS buffer (pH 7.0) prepared with deuterium oxide ($^2$H$_2$O). Incorporation of a single deuteron into both **7** and **8** was indicated by the presence of a +1 Da-shifted molecular ion peak at $m/z$ 326.7 ([M – H]$^-$) for both reduced products (Supplementary Fig. 12). In contrast, no changes in the molecular weights of **3**, **9**, **10** or **11** (Supplementary Fig. 12) were observed. Subsequently, **7**-$^2$H and **8**-$^2$H were isolated from scaled up reactions and analyzed by $^1$H and $^{13}$C NMR (Supplementary Figs. 13, 14 and Table 2). Comparison of the $^1$H NMR spectra of **7**-$^2$H and **7** revealed that the 3-H proton signal of **7** had vanished in **7**-$^2$H and meanwhile the 3-Me doublet of **7** had collapsed to a singlet in **7**-$^2$H (Fig. 3a), which located $^2$H at C-3 of **7**-$^2$H. In support of this assignment, the COSY correlation between the H-2 and H-3 protons observed in **7** was absent in **7**-$^2$H (Supplementary Figs. 4, 13). Likewise, a singlet was also observed for the 3-Me along with the absence of any COSY correlation between H-2 and H-3 in **8**-$^2$H (Fig. 3a; Supplementary Figs. 13, 14). Moreover, the $^{13}$C NMR spectra of **7**-$^2$H and **8**-$^2$H both showed broadening and reduced intensity of the signals from C-3, and $^2$H–$^{13}$C couplings at C-3 were also observed (Supplementary Figs. 13, 14)[40]. These observations demonstrate that a single solvent hydron is incorporated at C-3 during the reduction of **1** to both **7** and **8**.

When FST C (**1**) was incubated with NADH and FAD under $^{18}$O$_2$ (Fig. 3b, traces i−iii), LC-MS analysis of the reaction products revealed an increase of +2 Da in the molecular mass of the oxidized products yielding **3**-$^{18}$O, **9**-$^{18}$O, **10**-$^{18}$O and **11**-$^{18}$O (Fig. 3a; Supplementary Fig. 15). These data indicate that the conversion of **1** to **3**, **9**, **10**, and **11** involves incorporation of one oxygen atom from O$_2$. Furthermore, the relative production of **3/9** and **10/11** versus **7/8** was O$_2$ dependent, because incubation of FST C (**1**) with FAD and NADH after saturating the reaction buffer with O$_2$ led to a significant increase in the levels of the former (**3**, **9**–**11**) versus the latter (**7** and **8**); in contrast, saturating the reaction buffer with N$_2$ led to **7** and **8** as the dominant products (Fig. 3b, traces iii and iv). To locate the exact position of $^{18}$O incorporation during the reaction (**1 → 3/9**), a scaled up reaction was performed by incubating **1**, FAD, and NADH in $^{18}$O$_2$-saturated PBS buffer (pH 7.0). Two products **3**-$^{18}$O ([M – H]$^-$ $m/z$ 343.0718) and **9**-$^{18}$O ([M – H]$^-$ $m/z$ 343.0722) (Fig. 3a; Supplementary Fig. 16) were isolated. NMR analysis showed that the $^{18}$O label was incorporated at C-3 of **9**-$^{18}$O (Supplementary Fig. 17). Specifically, comparison of the $^{13}$C NMR data for **9** and its $^{18}$O isotopolog **9**-$^{18}$O demonstrated a 3.8 Hz upfield shift of $\delta_{C-3}$ in the latter relative to that of **9** (Supplementary Fig. 18 and Table 4), supporting the $^{18}$O label at C-3 (i.e., **9**-$^{18}$O)[41,42].

## Probing parameters affecting the epoxide ring opening reactions

The pH dependence of the FAD/NADH-dependent decomposition of **1** was subsequently investigated by performing the reaction in buffers over the pH range 3–10. Following the incubation of 100 μM **1** with 100 μM FAD and 10 mM NADH for 2 h at 30 °C, **1** was more efficiently converted to the epoxide ring-opened products with buffers in the pH range 5–7; however, the reaction was less efficient under more alkaline conditions at pHs 9–10 (Supplementary Fig. 19). FST B1 (**7**) was rather stable at pH 3–7; however, it readily converted to **8** at higher pH (Supplementary Fig. 20), where it also underwent oxidation to yield the previously reported natural product FST A (**12**) (Fig. 3b, trace v; Supplementary Fig. 20 and Table 1)[12]. FST B2 (**8**) was also stable at pH 3–6, and readily converted to **7** and **12** in more alkaline buffers (Supplementary Fig. 20). In contrast, FST C (**1**) was highly stable at pH 3–10, even in the presence of FAD or NADH alone (Supplementary Fig. 21) again demonstrating that decomposition of the epoxide in **1** requires the presence of both FAD and NADH.

A time course analysis of 100 μM **1** with 100 μM FAD and 10 mM NADH demonstrated the transient formation of an intermediary species **13** that vanished after longer incubation (Supplementary Fig. 22). A fixed initial concentration of **1** (100 μM) was also incubated with varying concentrations of FAD (1, 10, and 100 μM) and NADH (2, 5, and 10 mM) for 30 min at 30 °C (Supplementary Fig. 22). The lifetime of the transient intermediate was significantly extended at decreased levels of FAD (Fig. 3b, trace vi). This allowed collection of the intermediate **13** followed by immediate reinjection on the HPLC demonstrating its conversion to **7** and **12** (Fig. 3b, trace vii). When the isolated intermediate **13** was immediately incubated with FAD under O$_2$, the production of **7** and **12** was again observed (Fig. 3b, trace viii); however, if NADH was also included in the incubation, then **3** was instead observed as the major product with **12** as the minor product, and **7** was no longer observed (Fig. 3b, trace ix). The UV absorbance spectrum of the intermediate **13** was highly similar to those of **1**, **3**, and **7**, and LC-MS analysis of the intermediate indicated an $m/z$ 325.7 for the [M−H]$^-$ ion (Supplementary Fig. 23). Based on these measurements, the intermediate is tentatively proposed to be the enol **13** (Fig. 3b), which is essentially a tautomer of **7** and **8** and thus has the same oxidation state as these two products.

While several attempts were made to isolate the intermediate in order to confirm its structural assignment as **13**, only **7** was obtained as the major product along with another minor compound **14**. This is consistent with the poor stability of the intermediate and its facile conversion to **7** (Fig. 3b, trace vii, Supplementary Fig. 24). Nevertheless, **14** could be characterized and was thus identified as the C-3 peroxide designated hydroperoxyfluostatin (Fig. 3a; Supplementary Fig. 25 and Table 5); however, poor recovery of **14** hampered collection of sufficient spectra for a complete characterization of its absolute configuration at C-3. Hydroperoxyfluostatin (**14**) was stable in water, even in the presence of FAD (Fig. 3b, traces x and xi). While the coincubation of **14** with NADH alone led to multiple products **3**, **9**, **10** and **11**, the inclusion of FAD with NADH did not change the product profile (Fig. 3b, traces xii and xiii; Supplementary Fig. 26). Given the observed conversion of **14** to both **3** and **9**, compound **14** is likely a mixture of the (1$S$,2$S$,3$R$) and (1$S$,2$S$,3$S$) stereoisomers. The fact that isolation of **13** afforded a minor amount of **14** suggests that **14** is another intermediate that coelutes with **13** as opposed to being derived from **13**. Furthermore, while **3**/**9** could be generated from both **13** and **14**, **7**/**8** were only produced from **13**, whereas **10**/**11** were exclusively derived from **14** (Fig. 3a).

When other flavin-related cofactors were considered, FMN and riboflavin in place of FAD were also found to efficiently mediate the decomposition of **1** in the presence of NADH (Fig. 3b, traces xiv-xvi). Isoalloxazine and NADH were also able to facilitate the formation of **3** from **1**, albeit with a much lower efficiency (Fig. 3b, trace xvii). Only the

products **3**, **9**–**11** were observed using the cofactor F$_{420}$ in the presence of the reducing system constructed from glucose 6-phosphate and F$_{420}$-dependent glucose-6-phosphate dehydrogenase (FGD)[43–45], but notably no production of the reduced species **7** and **8** was observed (Fig. 3b, trace xviii). The observation of oxygen incorporation in the presence of F$_{420}$ and NADH was unexpected, because reduced 5-deazaflavin has generally be reported to react very slowly with molecular oxygen[46,47]. Nevertheless, the results with flavin cofactor compatibility shown in Fig. 3b (traces xiv-xviii) were found to be reproducible (Supplementary Fig. 27). NADPH was also found to be an effective cofactor similar to NADH and thus support the epoxide ring opening reactions (Supplementary Fig. 19).

## Generalization of the epoxide ring-opening reaction

To explore the generality of this reaction, an array of FST derivatives **15**–**21** (Fig. 4) were tested as possible oxirane reactants. Although FST F (**15**) is inert to Alp1U[18,33], it could be efficiently converted in the presence of FAD and NADH to 1-$O$-methyl-FST C2 (**22**), 1-$O$-methyl-FST B1 (**23**), 1-$O$-methyl-FST C3 (**24**), and 1-$O$-methyl-FST B2 (**25**) (Fig. 4 and Supplementary Fig. 28), which were structurally characterized by NMR and ECD spectroscopic analysis (Supplementary Tables 7, 8 and Figs. 29–32), as well as single crystal X-ray diffraction analysis (e.g. **23**, CCDCC 2036400, Supplementary Table 3).

Similarly, 7-$O$-methyl-FST C (**16**), 6-$O$-methyl-FST C (**17**), FST D (**18**), FST S (**19**), and difluostatin H (DiFST H, **20**) could also undergo epoxide-ring opening reactions upon treatment with FAD/NADH to produce a set of multiple products related to **3**, **7**, **8**, and **9** (Fig. 4, Supplementary Figs. 33–37). The major products from **16** and **17** were structurally characterized by 1D and 2D NMR as well as ECD spectroscopic analysis as FST M (**26**)[17], and 6-$O$-methyl-FST B1 (**27**), respectively (Fig. 4, Supplementary Table 9 and Figs. 38, 39). The structures of the other products from reactions with **18**–**20** were deduced based on LC-MS analyses. Notably, FST Q (**21**), which differs from **16** in terms of a C-4 hydroxyl instead of a C-4 carbonyl, was unreactive with FAD/NADH (Supplementary Fig. 40) highlighting the essential role of the C-4 carbonyl in this chemistry.

The epoxides **28**–**37** (Fig. 4) were also tested as putative oxirane substrates. No activities were observed with **28**–**35** (Supplementary Figs. 41–46)[16,40]. In contrast, treatment of auxarthrol H (**36**)[48] with FAD/NADH led to multiple products (Supplementary Fig. 47), one of which was determined to be **38**, which is a structural analog of auxarthrol F (Fig. 5; Supplementary Fig. 48 and Table 10)[49], by comparison with the experimental and calculated ECD spectra of **36** and **38** (Supplementary Fig. 49). Reaction with menadione 2,3-epoxide (**37**) yielded three products (Supplementary Table 11 and Figs. 50–53) including 3-hydroxy-3-methyl-2,3-dihydronaphthalene-1,4-dione (**39**), 2-hydroxy-3-methyl-1,4-naphthoquinone (**40**, phthiocol, characterized by X-ray diffraction, CCDC 2036401; Supplementary Table 12), and 3-methyl-naphthalene-1,4-dione (**41**).

Two common structural features shared by all compounds that can react with FAD/NADH to undergo the epoxide ring opening reactions include (i) the presence of a carbonyl adjacent to the epoxide and (ii) the presence of an aromatic moiety adjacent to the carbonyl/epoxide pair (Fig. 4). The absence of a carbonyl group neighboring the epoxide ring in **21** and **28**–**31** may account for the observed lack of reactivity. While **36** and **37** underwent ring opening in the presence of FAD/NADH, no reaction was observed for **32**–**35** despite the presence of an epoxide/carbonyl pair in these compounds. This implies that conjugation of the carbonyl/epoxide pair to an aromatic system is also necessary for ring opening of the epoxide.

The commercial compound vitamin K1 2,3-epoxide (**42**) was also tested as a putative substrate, which features both an epoxide/carbonyl pair and an adjacent aromatic ring. However, incubation of **42** with FAD/NADH led to no changes in either its TLC retention profile or spectroscopic properties implying no reaction (Supplementary

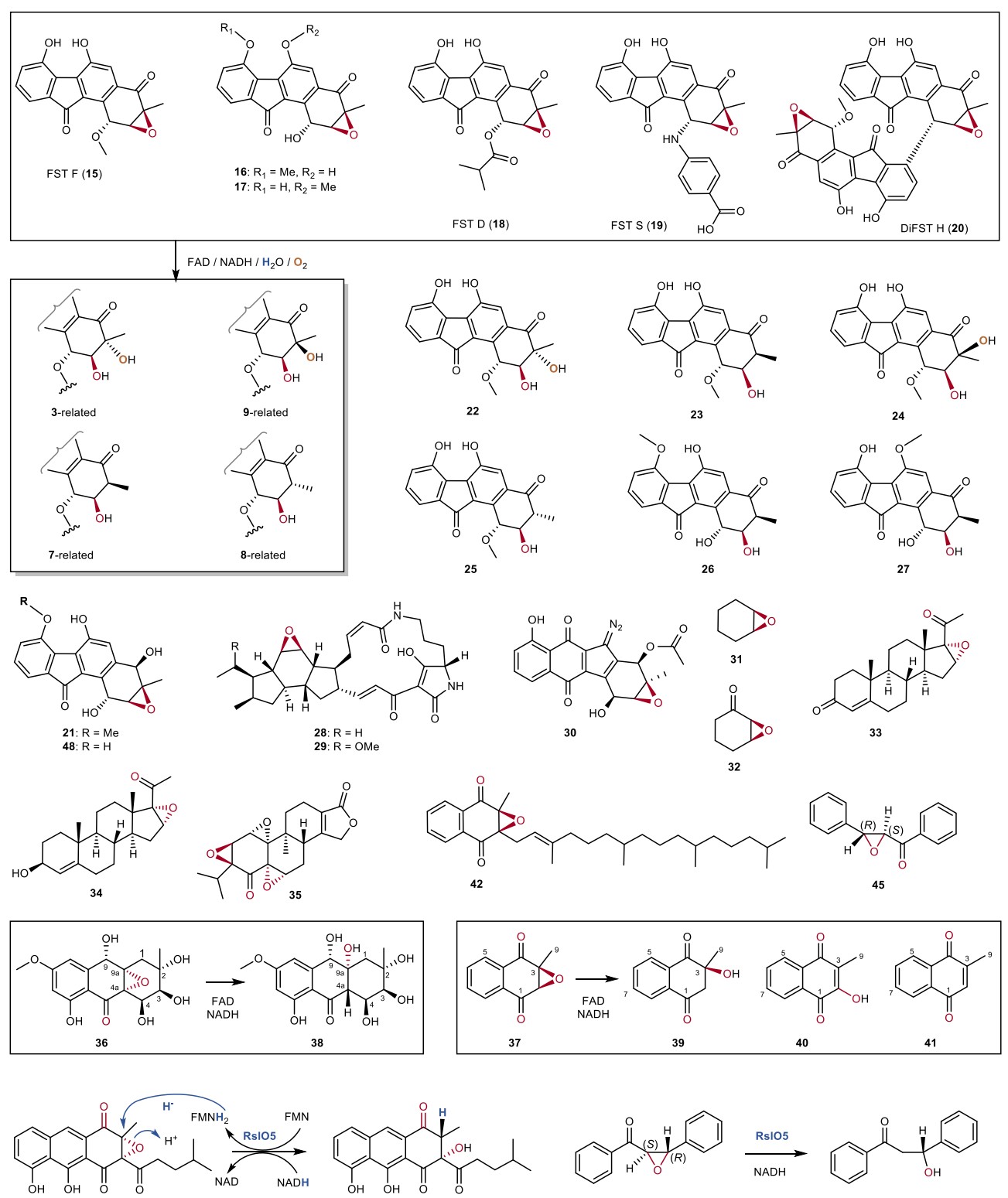

**Fig. 4 | Expanded substrate and product profile for the FAD/NADH-mediated epoxide ring opening reactions and comparison with the flavoenzyme RsIO5-catalyzed reactions.** FST derivatives **15–20** could undergo epoxide-ring opening reactions upon treatment with FAD/NADH to produce multiple products related to **3**, **7**, **8**, and **9**. FST Q (**21**) was unreactive with FAD/NADH. The epoxides **28–35** displayed no activities with FAD/NADH. Treatment of auxarthrol H (**36**) or menadione 2,3-epoxide (**37**) with FAD/NADH yielded multiple products. The epoxides **42–46** were found to be unreactive with FAD/NADH. The oxidoreductase RsIO5 catalyzed a reductive epoxide ring opening reaction on **43** to produce **44** during rishirilide biosynthesis and chalcone α,β-epoxide (**46**) was found to be a substrate of RlsO5 to yield the single product **47**.

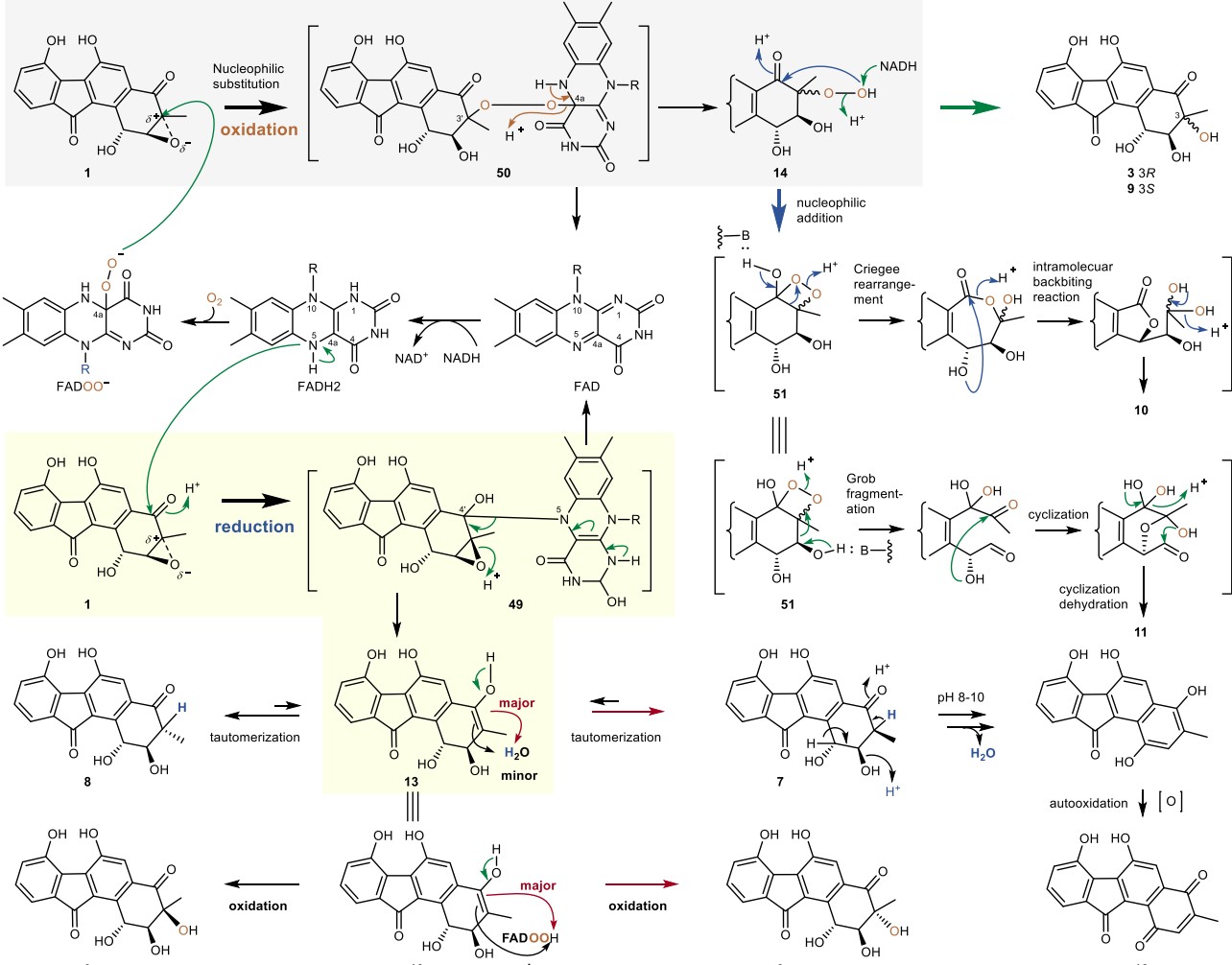

**Fig. 5 | Proposed mechanisms for flavin-enabled epoxide ring opening reactions.** The reaction can proceed via reductive and oxidative pathways leading to multiple products of different redox states depending on the presence of NADH and O2. The key transformations in the reductive reactions (yellow background) are proposed to involve a flavin N5–C4' adduct (**49**) to produce the enol intermediate

**13**, which can undergo tautomerization to yield **7** (major) or **8** (minor), and oxidation to produce **3** (major) or **9** (minor). **7** can be further autooxidized to **12**. Whereas the oxidative ring opening is likely to involve a flavin C4a–O–O–C3' adduct (**50**) to give the peroxylated intermediate **14** (gray background), which can undergo reduction to **3** and **9** by NADH, and further rearrangements leading to **10** and **11**.

Fig. 54). A recent study demonstrated that the flavoenzyme RslO5 catalyzed a hydride-mediated reductive epoxide ring opening reaction to convert **43** to **44** during rishirilide biosynthesis (Fig. 4)[11]. The reaction proceeded in a manner that was highly similar to the conversion of **1** to **7**. However, the production of **44** from **43** was confirmed to be strictly RslO5-dependent, and **43** was unreactive in the presence of FMN and NADH[11]. A possible explanation is that the presence of a long side chain next to the epoxide in both **42** and **43** might hinder both approach and orientation of the flavin thereby preventing suitable interactions for ring opening (Supplementary Fig. 55). Two additional commercial compounds *trans*-1,3-diphenyl-2,3-epoxypropan-1-one (**45**) and chalcone α,β-epoxide (**46**) were found to be unreactive with FAD/NADH; however, **46** but not **45** was accepted as a substrate by RlsO5 to yield the single product **47** (Supplementary Table 13 and Figs. 56, 57), which indicates that RslO5 is specific for only one of the two enantiomers. Deuterium incorporation into **47** was not observed from the RslO5 reaction in buffered $^2H_2O$ but was indeed observed when carrying out the RslO5 reaction in a coupled assay either with the glucose dehydrogenase from *Bacillus megaterium* DSM 2894 (BmGDH) to provide (*S*)-[4-$^2$H]NADH[50], or with the glucose dehydrogenase from *Thermoplasma acidophilum* ATCC 25905 (TaGDH) to provide (*R*)-[4-$^2$H]NADH[51], respectively, employing D-[1-$^2$H]glucose as

the deuterium donor (Supplementary Fig. 58)[52]. Previous studies demonstrated that the RslO5-H172N mutant had a deficiency in FMN binding and was inactive with the native substrate **43**[11], suggesting the critical role of FMN in the RslO5-catalyzed reaction. Taken together, a hydride from FMNH2 originating from NADH is proposed to be the effective nucleophile for RslO5-catalyzed epoxide opening reactions. In contrast, no deuterium incorporation into **7** was observed in reactions of **1** and FAD with either (*S*)-[4-$^2$H]NADH or (*R*)-[4-$^2$H]NADH generated as described above (Supplementary Figs. 58, 59). Therefore, nonenzymatic reduction of **1** to **7** in the presence of FAD/NADH differs from that of RslO5 catalysis with respect to the ultimate origin of the C-3 hydrogen.

## Discussion

Epoxide ring opening reactions are common and important in both biological processes and synthetic applications[1,2]. Nonenzymatic epoxide ring opening reactions are typically redox neutral proceeding through nucleophilic addition with the preferred regioselectivity and stereoselectivity contingent upon the nature of the epoxide and on the reaction conditions to generally yield products with a *trans*-vicinal diol[3,4]. Similar chemistry is encountered among the α/β-epoxide hydrolases that also generate *trans*-vicinal diol products[53]. For

example, the hydrolase AlpU catalyzes the epoxide hydrolysis of the FST C (**1**) epoxide to afford **2** and **3**, each of which exhibits a *trans*-vicinal diol (Fig. 1)[33]. In contrast, the flavoenzyme RslO5 was recently reported to catalyze a reductive epoxide ring opening of rishirilide thereby yielding a monohydroxylated product[11]. The RslO5 chemistry is thus distinctly different from that of the previously reported hydrolytic reactions and instead represents hydrogenation of an epoxide by reduced flavin, which is evident from the results using isotopically labeled substrates (Supplementary Fig. 58).

In this study, FAD/NADH is shown to induce the ring opening of suitably activated epoxides leading to a variety of products with different redox states. Under anaerobic conditions epoxide ring opening proceeds reductively in the presence of FAD/NADH via a mechanism that may be unrelated to that of RslO5 catalysis. The resulting monohydroxylated species, however, are highly susceptible to oxidation by molecular oxygen resulting in quinones (e.g., **12**) and vicinal alcohols in reactions that can also be accelerated by the presence of FAD. Under aerobic conditions, the epoxide ring opening appears to proceed through the formation of an α-ketoperoxylated species which can be reduced by NADH to again generate vicinal diols. Alternatively, the peroxylated intermediate can undergo further rearrangement leading to both ring contracted (e.g., the five-membered species **10**) as well as ring expanded (e.g., the seven-membered species **11**) oxidized products.

On the basis of the accumulated experimental evidence, a putative mechanism is proposed for the FAD/NADH enabled nonenzymatic epoxide-ring opening reactions (Fig. 5). Since both FAD and excess NADH are required for the observed epoxide ring opening reactions, NADH is proposed as the reductant with FAD serving as a redox catalyst. When the reaction is conducted aerobically, O₂ is proposed to act as the oxidant. Given that NADH alone could not initiate the reaction, the reduced flavin is proposed to be the species directly responsible for the reduction of **1**. It is possible that **7** is directly formed via a hydride transfer from a reduced flavin species (e.g., N5-H of FADH₂) to C3 of **1** thereby opening the epoxide ring, which would resemble the RslO5 catalyzed epoxide reduction reaction (Fig. 4). In this case, the observation that the product **7** has its H-3 derived from solvent could be explained by ready exchange of N5-H of the reduced flavin with solvent. This would imply that **7** should be the immediate reduction product prior to tautomerization to the observed intermediate **13**. However, a time course assay using 0.5 μM **1** with 10 μM FAD and 2 mM NADH demonstrated that **13** accumulates first followed by conversion to **7** and **8**. In contrast, no significant accumulation of **13** was observed in incubations of **7** under similar conditions (Supplementary Fig. 60). These results imply that **13** is the initial reduction product rather than **7** or **8**, and thus nonenzymatic reduction of **1** in the presence of FAD/NADH does not proceed via direct hydride transfer to C-3 as is proposed for RslO5 catalysis.

An alternative mechanism involves activation of the oxirane by the adjacent carbonyl coupled to an aromatic ring and undergoes reduction leading to an unstable intermediate. This intermediate is unlikely to be 7-demethyl-FST Q (**48**) (Fig. 4), which is the product of a direct C-4 carbonyl reduction in **1**, because the previously isolated FST Q (**21**)[17] is inert to the FAD/NADH reaction system (Supplementary Fig. 40). Instead, the intermediate during reduction of FST C (**1**) is hypothesized to be an enol species such as **13**, which is consistent with the observation that **13** readily tautomerizes to yield **7** and **8**. This mechanism is also consistent with the incorporation of deuterium at H-3 of **7** and **8** when the reaction is carried out in ²H₂O. Compound **13** is also expected to be susceptible to autoxidation to yield **3** and **9** in a manner that is sensitive to the presence of O₂, FAD, and NADH.

Formation of **13** may proceed via the addition of the nucleophilic N-5 of the reduced flavin to **1**. This model is supported by the fact that no formation of **7** was observed using the reduced F₄₂₀ cofactor, which

lacks the nucleophilic N-5 but is still able to serve as a hydride donor[54]. Nucleophilic addition of FADH₂ to **1** may occur at either C-3 or C-4. Given the presence of a sterically bulky CH₃ group at C-3 of **1**, C-4 is proposed to be the likely site of nucleophilic attack, which is consistent with the requirement for a carbonyl adjacent to the epoxide and thus the different reactivity of **16** (C-4 carbonyl) versus **21** (C-4 hydroxyl). Considering that N-5 in FADH₂ possesses a lone pair, and C-4 in **1** is a trigonal carbonyl carbon, the redox reaction is proposed to happen via an n→π* interaction. Specifically, the nucleophilic N-5 donates the lone-pair (n) electron density into the trigonal carbene center (empty π* orbital) along the Burgi–Dunitz trajectory[55]. Mixing of these orbitals is thermodynamically favorable resulting in a covalent adduct **49** (Fig. 5)[56]. Subsequent N-1 deprotonation of the flavin moiety in **49** leads to the breakage of the N₅–C₄' linkage and release of the oxidized flavin thereby completing the reduction of **1** to the enol intermediate **13** with a Δ³,⁴ double bond (Fig. 5).

Intermediate **13** is highly unstable and undergoes facile tautomerization to incorporate a solvent hydron at the α-carbon (C-3) either from the *si* face (the major route) or the *re* face (the minor route) of the sp² planar structure, to yield **7** or **8**, respectively[57]. This mechanism is consistent with the site-specific incorporation of deuterium at C-3 in **7** and **8** from buffered ²H₂O (Fig. 3a). Despite the 2,3-*cis* configuration of **7** compared to **8**, the former appears to be the thermodynamically favored tautomer with an equilibrium constant of approximately $[7]_{eq}/[8]_{eq} = 5.4$ (Supplementary Fig. 61). Moreover, **7**, **8** and the putative enol intermediate **13** are also susceptible to autooxidation to yield **12** (Supplementary Fig. 62).

The putative enol intermediate **13** also appears to be sensitive to molecular oxygen and can undergo autoxidation in the presence of FAD and NADH. Reduced flavin is known to react with O₂ to form the covalent Fl₄ₐₒₒₕ adduct (FAD → FADOOH)[58], and in some cases, H₂O₂ may be released from FADOOH to serve as the oxidant[37,59]. However, neither **1** nor **13** can react with H₂O₂ (Supplementary Figs. 24, 63). Thus, the intermediate **13** is proposed to react with the electrophilic C(4a)-hydroperoxy group of FADOOH to afford the C-3 monooxygenated products (see Supplementary Fig. 64). Hydroxylation can occur either from the *si* face (major route) or the *re* face (minor route) of the planar double bond (Δ³,⁴) to produce **3** or **9**, respectively, with concomitant release of the oxidized flavin (Fig. 5). The proposed mechanism is supported by the observations that (i) the O incorporated into both **3** and **9** at C-3 originates in O₂, and (ii) the conversion of **13** to **3** and **9** requires the presence of FAD, NADH and O₂.

Isolation of intermediate **14** indicates that **3** and **9** can also be produced in an alternative way that does not involve **13**. In the presence of molecular oxygen, the flavin C(4a)-peroxide (FADOO⁻) may serve as a nucleophile to regioselectively attack C-3 of **1** at either the *si* face (major route) or the *re* face (minor route) to open the epoxide ring, resulting in the putative adduct **50** (Fig. 5). The presence of the C-4 carbonyl oxygen with electron-withdrawing effect may cause changes of the electron density of C-3[60], to make C-3 a better site for nucleophile substitutions than C-2 of the epoxide in **1**. Deprotonation of the flavin N-5 in **50** breaks the C₄ₐ–O covalent bond to release **14** and FAD (Fig. 5). The product **14** is a mixture of C-3 isomers and easily reduced to **3** and **9** by NADH (Fig. 5, Supplementary Fig. 65). Coincubation of **14** with NADH also leads to **10** and **11**, which is proposed to involve a Baeyer-Villiger reaction-like mechanism. The C-3 hydroperoxide in **14** may thus perform an intramolecular nucleophilic addition at the C-4 carbonyl to form the putative intermediate **51**, which can partition to generate **10** and **11**. In the former case, intermediate **51** may undergo a Criegee rearrangement to yield a seven-membered lactone ring[61,62], followed by an intramolecular backbiting addition of the C-1 hydroxy at the C-4 carbonyl that leads to **10** after dehydration (Fig. 5)[63]. Alternatively, the 1,2-dioxetane unit of **51** may undergo heterolytic cleavage, followed by Grob-like fragmentation to yield an oxoheptanoate intermediate[64,65], which eventually generates the

hemiacetal **11** after successive intramolecular cyclizations and dehydration[66].

In conclusion, the reductive epoxide ring opening is suggested to proceed mechanistically via a transient adduct (**49**) with an $N_5$–$C_{4'}$ covalent bond that is formed between N-5 of the flavin cofactor and C-4 of the substrate FST C (**1**) (Fig. 5). While tautomerization of the resulting product can yield **7** and **8**, nucleophilic reaction with FADOOH can also afford the diol products **3** and **9**. Oxidative epoxide ring opening of **1** produces the intermediate **14** via a putative $C_{4a}$–O–O–$C_{3'}$ covalent adduct (**50**, Fig. 5). The canonical reactivity of flavin cofactors involves redox chemistry through N-5 hydride transfer or oxygen activation at C-4a in the majority of cases[67]. However, a few flavoenzymes have been reported to catalyze the formation of covalent flavin N-5–substrate adducts in a redox-neutral manner[68], with examples including the iminium-adducts in the catalytic cycle of UDP-galactopyranose mutase[69], alkyl-dihydroxyacetone phosphate synthase[70], flavin prenyltransferase[71], thymidylate synthase[72], and 2-haloacrylate hydratase[73].

The present study provides an example likely involving flavin N-5 adduct formation during flavin-catalyzed nonenzymatic reactions (Fig. 5). Furthermore, in addition to serving as a nucleophile in mediating hydroxylation, epoxidation and Baeyer-Villiger oxidation in flavoenzyme catalysis[58], the flavin C(4a)-peroxide (FADOO[−]) is proposed to also facilitate the formation of peroxylated intermediates such as **14** (Fig. 5). Flavin and flavin derivatives are well known to participate as cofactors in a wide variety of reactions found in nature[67,74]. Previous reports have shown that natural flavins can catalyze nonenzymatic oxidative decarboxylation of picolinic acid derivatives[75] and the nonenzymatic iodination of diverse aromatic compounds[37,59]. In this study, the chemistry of flavins is expanded to epoxide ring opening reactions in both a reductive and an oxidative manner to generate multiple ring-opened products. Consequently, the flavin chemistry identified in this study may be applicable to other epoxides and thus holds promise as a useful tool in organic synthesis.

## Methods

### Bacterial strains, compounds, and reagents
The strain *Micromonospora rosaria* SCSIO N160 was used for the isolation of FSTs[15]. FST Q (**21**) was previously isolated from *Streptomyces* sp. PKU-MA00045[17]. Chemicals, enzymes, and other molecular biological reagents were purchased from standard commercial sources and used according to the recommendations of the manufacturer.

### General analytical HPLC methods
HPLC analysis of reactions was generally carried out on an Agilent 1260 Infinity series instrument (Agilent Technologies Inc., USA) using a reversed phase C18 column (Kinetex® 5 μm C18 100 Å, LC Column 150 × 4.6 mm, Phenomenex, USA) or a polar column (Comixsep®, P/N FMG-BPF5-EONU, Polar BiPFP 5 u, 250 × 4.6 mm, China) with UV detection at 256 or 304 nm under the following program: solvent A, 10% acetonitrile (MeCN) in water supplementing with 0.1% formic acid; solvent B, 90% MeCN in water; 5% B to 80% B (0–20 min), 80% B to 100% B (20–21 min), 100% B (21–24 min), 100% B to 5% B (24–25 min), 5% B (25–30 min); flow rate at 1 mL min⁻¹.

### Expression, purification of recombinant enzymes, and general in vitro enzyme assays
The DNA fragment of *rslO5* (GenBank: KJ437438.1 [https://www.ncbi.nlm.nih.gov/nuccore/KJ437438.1]) from the rishirilide biosynthetic gene cluster in *Streptomyces bottropensis*[11] was synthesized and cloned into pET28a to afford the plasmid pCSG8112 (Supplementary Table S14). When the cultures of *E. coli* BL21(DE3)/pCSG8112 in LB media containing kanamycin (50 μg mL⁻¹) were grown to an $OD_{600}$ of about 0.6 at 37 °C, the production of RslO5 was induced by the addition of isopropyl-β-D-thiogalactopyranoside (IPTG) to a final concentration of 0.1 mM. The cultures were grown at 16 °C for an additional 12 h. The cells were then collected by centrifugation and were resuspended in the lysis buffer (20 mM Tris-Cl, 500 mM NaCl, and 5 mM imidazole, pH 8.0) for sonication. Purification of *N*-His₆-tagged RslO5 was conducted using Ni-NTA affinity chromatography according to the manufacturer's manual (Novagen, USA). After desalting with PD-10 column (GE Healthcare, USA), the purified RslO5 was stored in the storage buffer (10% glycerol, 1 mM DTT, 50 mM Tris-Cl, 100 mM NaCl, pH 8.0) at −80 °C for further use. The expression and purification of Fre[34], Alp1U[18], FlsH[18], XiaK[38], TiaM[36], and FlsO2[35] were performed similar as described for RslO5. The standard in vitro enzyme assays (Alp1U, FlsH, Fre, XiaK, TiaM, Fre, or FlsO2) contained 100 μM FST C (**1**) and 10 μM of purified recombinant enzyme in 50 mM PBS (phosphate buffered saline) buffer (pH 7.0) in a total volume of 100 μL with incubation at 30 °C for 0.5–2 h. The reactions were stopped by adding 100 μL of ice-cold MeOH and were monitored using the general analytical HPLC methods.

### Nonenzymatic reactions with flavins
A typical reaction mixture included 100 μM **1**, 100 μM FAD and 10 mM NADH in 50 mM PBS buffer (pH 7.0) in a total volume 100 μL with incubation at 30 °C for 0.5–2 h. In general, the reactions were terminated by adding 100 μL of ice-cold MeOH and analyzed by HPLC using the general analytical HPLC method or subjected to LC-MS analysis. For reactions under anaerobic conditions, the reaction buffers were flushed in $N_2$ for 5 min before adding **1**, cap-sealed using Parafilm and flushed with $N_2$ during incubation at 30 °C for 30 min. For reactions under aerobic conditions, the reaction buffers were flushed in $O_2$ for 5 min before adding **1**, cap-sealed using Parafilm and flushed with $O_2$ during incubation at 30 °C for 30 min. For isotope labeling reactions with ²H, each reaction was carried out in 50 mM PBS buffer (pH 7.0) prepared with deuterated water (²H₂O). For isotope labeling reactions with ¹⁸O₂, a reaction mixture containing 100 μM FAD and 10 mM NADH in 50 mM PBS buffer (pH 7.0) was degassed and flushed with ¹⁸O₂ for 5 min; after the addition of 100 μM **1**, ¹⁸O₂ gas was continuously bubbled into the reaction mixture for 2 h at room temperature with occasional shaking.

For determining the pH effects on the reactions, the assays were performed in a total volume of 100 μL containing 100 μM **1**, 100 μM FAD and 10 mM NADH in 50 mM buffers with pH values ranging from 3 to 10 with incubation at 30 °C for 2 h. The buffers were prepared as follows: citric acid/Na₂HPO₄ buffer (pH 3–6); PBS buffer (pH 7); boric acid/borax buffer (pH 8–9), borax/NaOH buffer (pH 10). As controls, 100 μM **1** was incubated in different pH buffers (pH 3–10) in the presence of 100 μM FAD alone (or 10 mM NADH alone) at 30 °C for 2 h.

For time course assay, reactions were performed by incubating 100 μM **1**, 100 μM FAD, and 10 mM NADH in 50 mM PBS buffer (pH 7.0) at 30 °C with sampling at different time points of 0, 2, 4, 6, 8, 10, 12, 14, 16, 20, 25, and 30 min. For assaying the effects of different FAD/NADH concentrations, reactions were carried out with 100 μM **1** in 50 mM PBS buffer (pH 7.0) with varying concentrations of FAD and NADH. The first set of reactions were conducted with 100 μM **1**, 1 μM FAD, and 2 (or 5 or 10 mM) NADH. The second set of reactions were performed with 100 μM **1**, 10 μM FAD, and 2 (or 5 or 10) mM NADH. The third set of reactions were carried out with 100 μM **1**, 100 μM FAD, and 2 (or 5 or 10) mM NADH. The reaction mixtures were incubated in 50 mM PBS buffer (pH 7.0) at 30 °C for 30 min.

To test cofactor compatibility, assays containing 100 μM of **1**, 100 μM FAD (or FMN/riboflavin/isoalloxazine) and 10 mM NADH in 50 mM PBS buffer (pH 7.0) in a total volume of 100 μL were incubated at 30 °C for 30 min. For testing F420 as a cofactor, assays containing 100 μM of **1**, 200 μM F420, 2.5 mM glucose-6-phosphate, 10 μM F₄₂₀-dependent glucose-6-phosphate dehydrogenase (FGD) in 50 mM Tris-HCl buffer (pH 7.5) in a total volume of 100 μL were incubated at 30 °C for 1 h.

The FAD/NADH-mediated epoxide ring opening reactions were tested with various substrates including both FST-related substrates **15–21** and FST-nonrelated substrates **28–37**, **42**, **45**, and **46**. A typical reaction mixture contains 100 µM substrate (**15–21**, **28–37**, **42**, **45**, or **46**), 100 µM FAD and 10 mM NADH in 50 mM PBS buffer (pH 7.0) in a total volume of 100 µL.

To test the reactivity of $H_2O_2$ with FST C (**1**), the assays were performed in a total volume of 100 µL containing 100 µM **1** and 30% $H_2O_2$ in 50 mM buffers at pH 3.0 (or pH 7.0 or pH 10.0) by incubation at 30 °C for 2 h.

### Preparation and isolation of 3, 7–11, 22–27 and 38–41

A scaled up reaction of **1** in a total volume of 1 L was performed in 50 mM PBS buffer (pH 7.0) containing 100 µM **1**, 10 mM NADH and 100 µM FAD at 30 °C for 2 h with occasional shaking. The reaction was quenched by the addition of an equal volume of ice-cold butanone and centrifuged at 2057 × *g* for 20 min at 4 °C. The reaction mixtures were then extracted with an equal volume of butanone three times and the solvents were removed under vacuum on an ice bath. The crude extracts were dissolved in 1.5 mL MeOH and subjected to semi-preparative HPLC using an Agilent Eclipse XDB-C18 column (250 mm × 9.4 mm, 5 µm; Agilent technology Ltd., USA) with an isocratic elution gradient of 70% A ($H_2O$ with 0.8% formic acid) and 30% B (MeCN) at a flow rate of 2.5 mL min⁻¹. In this way, compounds **3** (8.5 mg, 22.0%), **7** (22.0 mg, 57.0%), **8** (6.0 mg, 15.5%), **9** (2.6 mg, 6.7%), **10** (1.6 mg, 4.1%) and **11** (3.4 mg, 8.8%) were obtained. Similarly, a 40 mL scale of reaction with 100 µM FST F (**2**) afforded **22** (2.0 mg, 15%), **23** (6.0 mg, 46.0%), **24** (0.9 mg, 7.0%), and **25** (1.2 mg, 9.0%). The 30 mL scale reactions of **16**, **17** or **36** yielded **26** (1.3 mg, 44.0%), **27** (1.5 mg, 46.0%), or **38** (1.3 mg, 67.0%), respectively. A 30 mL scale reaction of **37** yielded **39** (1.8 mg, 11.0%), **40** (4.8 mg, 30.0%) and **41** (4.3 mg, 27.0%).

### Isolation of isotope-labeled products

For the isolation of **3**-¹⁸O and **9**-¹⁸O, a 400 mL scale reaction of **1** was carried out under ¹⁸O₂, consisting of 100 µM **1**, 100 µM FAD, and 10 mM NADH in 50 mM PBS buffer (pH 7.0) by incubation at 30 °C for 2 h. The reaction products were purified via semi-preparative HPLC as described above to afford **3**-¹⁸O (3.0 mg, 33.0%) and **9**-¹⁸O (1.2 mg, 16.0%). For the isolation of **7**-²H and **8**-²H, a 400 mL scale reaction of isotope labeling with ²H, was set up in 50 mM PBS buffer (pH 7.0) prepared with buffered ²H₂O, containing 100 µM **1**, 100 µM FAD, and 10 mM NADH, by incubation at 30 °C for 2 h. The reaction products were purified via semi-preparative HPLC to afford **7**-²H (11.5 mg, 65.0%) and **8**-²H (3.2 mg, 19.0%).

### The analysis of the intermediate 13

A reaction mixture containing 100 µM **1**, 10 µM FAD, and 2 mM NADH in 50 mM PBS buffer (pH 7.0) in a total volume 100 µL was incubated under a normal atmosphere at 30 °C for 20 min. The reactions were quenched by adding 100 µL of ice-cold MeOH. To check the stability of **13**, the intermediate **13** was collected from a general analytical HPLC run and was immediately reinjected for HPLC analysis. Similarly, to monitor the changes of **13** under various conditions, the freshly collected **13** was immediately incubated in 50 mM PBS buffer (pH 7.0) containing (i) 10 µM FAD; (ii) 30% $H_2O_2$; (iii) 2 mM NADH; (iv) 10 µM FAD and 2 mM NADH (in presence of excess of $O_2$) at 30 °C for 30 min. The reactions were quenched by adding 100 µL of ice cold MeOH and monitored by the general analytical HPLC method using a C18 column as well as a polar column.

### Isolation and stability of 14

Scaled up reactions in a total volume of 50 mL (100 µL × 500 Eppendorf tubes) were performed containing 100 µM **1**, 10 µM FAD, and 2 mM NADH in 50 mM PBS buffer (pH 7.0) at 30 °C for 20 min with the aim of isolating **13** for structure elucidation. The reactions in each Eppendorf tube were stopped by extracting three times with 100 µL of ice-cold butanone and the solvents were removed under vacuum on an ice bath. The crude extracts were then collected and dissolved in 1.0 mL MeOH and subjected to general analytical HPLC using a polar column. The collections were kept at −80 °C prior to lyophilization. The collected fractions were purified a second time to remove **7**, which had been generated from the decomposition of **13**. This process finally afforded a pure and stable compound (0.8 mg), which was dissolved in DMSO-$d_6$ for NMR analysis and confirmed to be **14**.

A set of assays were performed to check the stability of **14**. A typical reaction mixture included 100 µM **14**, 10 µM FAD, and 2 mM NADH in 50 mM PBS buffer (pH 7.0) in a total volume 100 µL. Control reactions were run in parallel by omitting FAD or NADH in the assays. Co-incubation of 100 µM **14** in 30% $H_2O_2$ were also performed in a total volume of 100 µL. The reaction mixtures were incubated at 30 °C for 30 min. The reactions were quenched after adding 100 µL of ice-cold MeOH and were analyzed by HPLC using a polar column following the general analytical HPLC method.

### In vitro RslO5 assays, isolation of 47 and deuterium incorporation studies

A standard RslO5 in vitro reaction mixture contained 100 µM *trans*-1,3-diphenyl-2,3-epoxypropan-1-one (**45**) or chalcone α,β-epoxide (**46**), 10 µM RslO5 and 5 mM NADH in 50 mM PBS buffer (pH 7.0) in a total volume of 100 µL and incubated at 30 °C for 2 h. The reaction with **46** was also performed in 50 mM PBS buffer (pH 7.0) prepared with deuterated water (²H₂O) and subjected to LC-HRESIMS analysis to investigate ²H incorporation into **47** from the solvent. To isolate **47** for structural elucidation, 15 mL reactions were performed in 50 mM PBS buffer (pH 7.0) containing 100 µM **46**, 10 µM RslO5 and 5 mM NADH at 30 °C for 6 h. The reaction was quenched with an equal volume of pre-chilled butanone. After centrifugation at 2057 × *g* for 20 min at 4 °C, the supernatants were extracted three times with an equal volume of pre-chilled butanone. The butanone was removed under vacuum on an ice bath. The crude extracts were dissolved in 500 µL of MeOH and were purified by semi-preparative HPLC using a C18 column (250 mm × 10.0 mm, 5 µm; Phenomenex, USA), with an isocratic elution gradient of 60% A ($H_2O$ with 0.8% formic acid)/40% B (MeCN) at a flow rate of 2.5 mL min⁻¹ to yield **47** (3.0 mg, 41.0%).

The purification of D-glucose dehydrogenases BmGDH from *Bacillus megaterium* DSM 2894 in *E. coli* BL21(DE3)/pRSF-BmGDH and TaGDH from *Thermoplasma acidophilum* in *E. coli* BL21(DE3)/pCSG3622, was carried out according to our previous study[76]. In case of the RslO5/GDH coupling assays, the reduction of NADP to NADPH was first performed in a 100 µL reaction containing 10 µM TaGDH (or BmGDH), 2 mM NADP and 100 mM [1-²H]D-glucose in 50 mM sodium phosphate buffer (pH 7.0) by incubation for 1 h at 37 °C for BmGDH or 50 °C for TaGDH, followed by the addition of 100 µM **46** and 10 mM RslO5 and further incubation at 30 °C for 6 h. The in situ generated ²H-labeled NADH was also reacted with **1** and FAD to monitor the incorporation of ²H in to the product **7**. The reactions were quenched with an equal volume of ice-cold MeOH and analyzed by LC-HRESIMS using the general analytical HPLC method.

### Quantum chemistry calculations

Mixed torsional/low-mode conformational searches were carried out by means of the Macromodel 10.8.011 software using the MMFF with an implicit solvent model for CHCl₃ applying a 21 kJ/mol energy window[77]. Geometry re-optimizations of the resultant conformers were carried out at the ωB97X/TZVP level with the PCM solvent model for MeCN. TDDFT ECD calculations were performed with Gaussian 09 for ECD using various functionals (B3LYP, BH&HLYP, CAM-B3LYP, PBE0) and the TZVP basis set with the same solvent model as in the preceding DFT optimization step[78]. ECD spectra were generated as the sum of Gaussians with 3000 and 3600 cm⁻¹ half-height widths, using

dipole-velocity-computed rotational strength values[79]. Boltzmann distributions were estimated from the ωB97X energies. The MOLEKEL software package was used for visualization of the results[80].

## X-ray crystallographic analysis

A single crystal of **7**, **10**, and **23** was obtained in a mixed solvent of MeOH and water, and single crystals of **40** were obtained in a mixed solvent of $CH_3CN$ and $H_2O$. The suitable crystals were selected, and the crystal data were recorded on an XtaLAB AFC12 (RINC): Kappa single diffractometer, with Cu Kα radiation ($\lambda = 1.541\,84$ Å). The crystals were kept at 99.9(7) K for **7**, 99.8(9) K for **23**, 100.00(10) K for **10** and 100.01(12) K for **40** for during data collection. Using Olex2[81], the structure was solved with the ShelXT[82] structure solution program using intrinsic phasing and refined with the ShelXL refinement package using least squares minimization.

## Data availability

The data and code in this manuscript have been appropriately deposited into a public data repository or provided in the supplementary information for public availability. Crystallographic data can be obtained free of charge for **7** (CCDC 2036399), **10** (CCDC 2129081), **23** (CCDC 2036400), and **40** (CCDC 2036401) via www.ccdc.cam.ac.uk/data_request/cif, or by emailing data_request@ccdc.cam.ac.uk, or by contacting The Cambridge Crystallographic Data Centre, 12 Union Road, Cambridge CB2 1EZ, UK; fax: +44 1223 336033. The GenBank accession number for DNA fragment of *rslO5* is KJ437438.1 (https://www.ncbi.nlm.nih.gov/nuccore/KJ437438.1).

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

## Acknowledgements

This work is supported in part by the National Natural Science Foundation of China (31820103003 to C.Z., 42176127 to W.Z., 41676165 to W.Z., 31630004 to C.Z., and 31700042 to C.Y.); Key Science and Technology Project of Hainan Province (ZDKJ202018 to C.Z.); MOST (2018YFA0901903 to Y.Z.); K.C. Wong Education Foundation (GJTD-2020-12 to C.Z.); the Guangdong Provincial Special Fund for Marine Economic Development Project (GDNRC[2021] 48 to C.Z.); Southern Marine Science and Engineering Guangdong Laboratory (Guangzhou) (GML2019ZD0406 to C.Z.); Youth Innovation Promotion Association CAS (2022349 to C. Y.) and the Science and Technology Planning Project of Guangzhou (202102020471 to C.Y.). B.C.D. acknowledges the support of CAS-TWAS President's PhD Fellowship. H.-w.L. acknowledges the support of Welch Foundation (F-1511). G.B. is supported by a Sir Charles Hercus Fellowship through the Health Research Council of New Zealand. We appreciate Professor Keqiang Fan from Institute of Microbiology, Chinese Academy of Sciences, for the generous gift of compound **30**. We would like to thank Professor Tadhg Begley of Texas A&M University for useful discussion. We are grateful to Dr. Z. H. Xiao, X. H. Zheng, A. J. Sun, Y. Zhang, and X. Ma in the equipment public service center at SCSIO for recording spectroscopic data. We also thank the data archive support from the National Earth System Data Center, National Science & Technology Infrastructure of China (http:// www.geodata.cn).

## Author contributions

B.C.D., W.Z., C.Y., H.C., L.P., and W.L. performed epoxide opening reactions, compound isolation, and structure determination. A.M. and T.K. conducted the ECD calculations. M.M. and G.B. contributed key compound and reagents. B.C.D., W.Z., M.W.R., Y.Z., T.K., H.w.L., and C.Z. analyzed the data and wrote the manuscript. C.Z. and H.w.L. directed the research. B.C.D. and W.Z. contributed equally to this work.

## Competing interests

The authors declare no competing interests.
