## [Peer Review File · Nature Communications]

REVIEWER COMMENTS

Reviewer #1 (Remarks to the Author):

In this work, the Zhang and Liu groups reported two novel types of Flavin mediated non-enzymatic epoxide ring opening reactions. They established that Flavin and NADH together can catalyze both the reductive and oxidative epoxide ring opening reactions of fluostatin C (FST C) to produce multiple FST analogues with different redox states. Through structural characterization of the products and careful analysis of the 2H and 18O incorporation in intermediates, they elucidated the catalytic mechanisms of the both reactions. They also verified that the presence of an adjacent carbonyl coupled to an aromatic moiety are required for the two reactions and the both reactions can be applied to some other natural products with similar structural feature. This work discovered two important epoxide-opening reactions which are distinct from the previous known reactions in natural products biosynthesis. It represents a significant addition to the organic chemistry, natural products chemistry and flavin chemistry, and could be useful for natural products combinatorial biosynthesis. Overall, the quality of the study is very high, and I have only minor comments and suggestions. Otherwise, I think the paper is well suitable for publication in Nature Communications.

1. Is NADPH also effective to support the reactions?
 2. Does the FST-producing strain produce compounds 7-11? Did the authors check the fermentation broth? It will be better to include a short discussion about the in vivo results.
 3. It will be better to provide experimental data about the key flavin-FST adducts, although the proposed catalytic mechanisms are pretty reasonable. I think UV-Vis analysis will be helpful to get the proof.
2. Last line in page 4: Fig. 2a, traces iv-vi should be Fig. 2a, trace vi.

Reviewer #2 (Remarks to the Author):

Biosynthetic pathways often take advantage of intrinsic chemical reactivity of building blocks and intermediates. Here in vitro interaction of cofactors, flavin and NADH, with highly functionalized and reactive keto-epoxide substrates are examined. A great deal of careful isolation and structural characterization of multiple products is described and finally unifying mechanistic interpretation is applied to rationalize their formation. Effects of pH, D2O labeling, 18O labeling, time-course

experiments, presence and absence of flavin and/or NADH are examined. All of these experiments are extensively documented and the text itself is very well referenced.

While Nat Commun, unlike some journals, solicits specialized papers in its wide subject areas. This would be such a paper. It is a very detailed analysis of a very particular natural product structural type. Biosynthetic proteins are specifically not examined, but inherent cofactor reactivity with those substrates is separately examined. I am willing to go along with the mechanisms proposed and the overall quality of the work is high. I hesitate, however, that the products obtained are not unexpected from these cofactors and the narrow, albeit correctly done, impact of the work is not up to the standard of Nat. Commun.

Reviewer #3 (Remarks to the Author):

The research presented by the Zhang group in the underlying publication sets a very nice example, how thorough a basic and already well investigated reaction like regioselective epoxide opening can be investigated. It beautifully showcases the ability of flavin cofactors to react with a broad variety of epoxides, without the need of a catalysing enzyme, but still selecting for specific substitution patterns contributing to the correct steric and electronic environment. While the reaction unfortunately does not follow one single reaction path but leads to up to 6 different outcomes, the authors nicely illustrate the different pathways and do a great job on the structure elucidation of the large number of products they isolated. The products with expanded and contracted ring structures seem especially interesting to me, since those would not be easily achievable with established synthetic methods. Another very interesting finding is the peroxide 14, and, although a stereochemical assignment of carbon number 3 would be nice to have, the existence of this intermediate and the prove of the further conversion to only a selection of the originally observed products arising from the FST C already gives it a very important role in the final mechanistic proposal. While the method as presented herein might not be of immediate use for the field of epoxide opening, a very important reaction in natural product total synthesis, it will surely inspire other groups to investigate the use of flavin cofactors in combination with (chiral) catalysts to direct it into one specific direction.

Given this inspirational aspect and the extremely thorough and detailed work provided by the authors I recommend this publication to be accepted if some changes are made to present this on an even better and journal-appropriate level:

1. While absolute amounts are given for the isolation of all compounds the yield is missing. This needs to be corrected.

2. Regarding the conversion of compound 42: did the authors consider different analytical techniques than TLC? While this is a very sensitive method, a spot on the same height, doesn't necessarily mean, that this is the same compound. Either a third lane with the putative product would be needed, or some other evidence in the form of a spectral analysis should be given.

3. Presentation of NMR data (a-h):

a. a lot of the NOESY data are not appropriately zoomed, that should be changed and the crosspeaks that were used for structure elucidation should be highlighted for an easier overview. Also, the NOESY spectrum of 10 is missing the structure of the associated compound.

b. Some of the 1D spectra and the axial spectra on the 2D spectra are not zoomed to an appropriate level as well, for better interpretation by the reader that should be changed. Also, a baseline correction should be done for all spectra to make them look more appealing.

c. The NMR spectra for compound 9 look surprisingly bad, considering that 2.6 mg of this compound have been isolated, while other spectra for smaller amounts look much better. A clean spectrum of the compound would be nice.

d. For the deuteration experiments on compounds 7 and 8 it would be nice to show the ^{13}C NMR spectra as additional proof for attachment of deuterium to carbon 3.

e. For compound 14 additional spectra would be nice, if that is possible. If not, due to the instable nature a short comment would be great.

f. The ^1H -NMR of compound 25 is missing integrals

g. In some of the NOESY spectra the hydrogen at C1 is drawn with a relative instead of an absolute configuration, which should be adjusted.

h. The NOESY for compound 47 is missing. Although it is not very helpful for structure elucidation, it should still be presented as part of a complete dataset.

4. For all newly identified compounds the specific rotation should be measured and reported, as part of a full characterisation for a chiral compound.

5. For the conversion of compounds 45 and 46 it would be appropriate to briefly mention, that those are enantiomers and that it's hence not surprising, that only one is converted by a chiral catalyst, i.e., the enzyme RslO5.

6. The mass spectra shown in the supporting information could be polished a little bit, m/z should be given in the appropriate form with "m" and "z" being in italics.

7. Lastly, the manuscript gets very lengthy in parts, where methodologies for structure elucidation are described very precisely in the main text. Given the fact that this is supposed to be a communication, a more concise presentation would be appropriate, and some parts should be shortened to fit the manuscript into the 5000 word limit which is slightly exceeded.

Apart from these, mostly cosmetic, corrections I found the manuscript very appealing to read (especially with the extraordinarily well organised and presented Figures!) and I think that it is of interest to a broad readership, from a biological, as well as from a chemical background.

Response to Reviewer Comments

Reviewer #1 (Remarks to the Author):

Comments:

In this work, the Zhang and Liu groups reported two novel types of Flavin mediated non-enzymatic epoxide ring opening reactions. They established that Flavin and NADH together can catalyze both the reductive and oxidative epoxide ring opening reactions of fluostatin C (FST C) to produce multiple FST analogues with different redox states. Through structural characterization of the products and careful analysis of the ^2H and ^{18}O incorporation in intermediates, they elucidated the catalytic mechanisms of the both reactions. They also verified that the presence of an adjacent carbonyl coupled to an aromatic moiety are required for the two reactions and the both reactions can be applied to some other natural products with similar structural feature. This work discovered two important epoxide-opening reactions which are distinct from the previous known reactions in natural products biosynthesis. It represents a significant addition to the organic chemistry, natural products chemistry and flavin chemistry, and could be useful for natural products combinatorial biosynthesis. Overall, the quality of the study is very high, and I have only minor comments and suggestions. Otherwise, I think the paper is well suitable for publication in Nature Communications.

Response 1. We appreciate the positive comments from reviewer #1.

Comment 1.1. *Is NADPH also effective to support the reactions?*

Response 1.1. NADPH, like NADH, is also an effective cofactor and can support the epoxide ring opening reactions. Thus, the same set of products are also observed upon incubation of FST C (**1**) with FAD and NADPH in the absence of any enzymes. This information has been included in the revised manuscript (see lines 204–205) as well as the **Supplementary Fig. 19**, which is reproduced below for convenience.

Supplementary Fig. 19b.

Comment 1.2. Does the FST-producing strain produce compounds 7-11? Did the authors check the fermentation broth? It will be better to include a short discussion about the *in vivo* results.

Response 1.2. The present study does not include an *in vivo* study of FST biosynthesis; however, this is an important question and a motivating factor behind our work. As such, a short description of naturally isolated FSTs had been included in the introduction (see lines 68–69). FST B has been previously isolated from *Streptomyces* sp. TA-3391 and *Streptomyces* strain Acta 1383 (refs. 12 & 13), and while its stereochemistry was not reported, it does have the same planar structure as 7 or 8. We have added a note to the manuscript (see lines 106–108) that explicitly points this out. In contrast, we are not aware of any published reports that compounds 9–11 have been isolated from a natural source, and we have noted this in the revised manuscript as well (see lines 126–127).

Comment 1.3. It will be better to provide experimental data about the key flavin-FST adducts, although the proposed catalytic mechanisms are pretty reasonable. I think UV-Vis analysis will be helpful to get the proof.

Response 1.3. We agree that additional experimental data characterizing the putative flavin-FST adducts would help to further test the proposed mechanisms, and several attempts were indeed made to do so. For example, efforts were made to determine the

molecular weight of the key flavin-FST adducts and record the UV absorbance of species produced at different reaction time points especially in the 1–10 min time range. Unfortunately, these efforts were unsuccessful likely due to the predicted instability of the putative flavin-FST adducts leading to their rapid decomposition. Nevertheless, our current data are consistent with the proposed mechanisms, which we agree will require additional work to test more fully in future works.

Comment 1.4. Last line in page 4: Fig. 2a, traces iv-vi should be Fig. 2a, trace vi.

Response 1.4. We have revised the text as suggested.

Reviewer #2 (Remarks to the Author):

Comment:

Biosynthetic pathways often take advantage of intrinsic chemical reactivity of building blocks and intermediates. Here in vitro interaction of cofactors, flavin and NADH, with highly functionalized and reactive keto-epoxide substrates are examined. A great deal of careful isolation and structural characterization of multiple products is described and finally unifying mechanistic interpretation is applied to rationalize their formation. Effects of pH, D₂O labeling, ¹⁸O labeling, time-course experiments, presence and absence of flavin and/or NADH are examined. All of these experiments are extensively documented and the text itself is very well referenced.

While Nat Commun, unlike some journals, solicits specialized papers in its wide subject areas. This would be such a paper. It is a very detailed analysis of a very particular natural product structural type. Biosynthetic proteins are specifically not examined, but inherent cofactor reactivity with those substrates is separately examined. I am willing to go along with the mechanisms proposed and the overall quality of the work is high. I hesitate, however, that the products obtained are not unexpected from these cofactors and the narrow, albeit correctly done, impact of the work is not up to the standard of Nat. Commun.

Response 2. We would like to thank reviewer #2 for the critical appraisal of our work. To the best of our knowledge, this is the first study to show that flavin cofactors can independently catalyze reductive and oxidative epoxide ring opening reactions. Furthermore, we have compared these reactions with the reductive epoxide ring opening reactions catalyzed by the oxidoreductase RslO5 and in doing so provided evidence that the mechanisms are

distinct. We have also demonstrated that the FAD/NADH mediated epoxide ring opening reactions are not limited to FSTs and can be generalized to other compounds featuring an epoxide adjacent to a carbonyl group that is conjugated to an aromatic moiety. Consequently, we believe this work will be of broad interest to the readership of *Nature Communications*, because it extends the repertoire of known flavin chemistry and provides new insights into natural product biosynthesis and a foundation for the development of new methodologies in organic synthesis.

Reviewer #3 (Remarks to the Author):

Comments:

*The research presented by the Zhang group in the underlying publication sets a very nice example, how thorough a basic and already well investigated reaction like regioselective epoxide opening can be investigated. It beautifully showcases the ability of flavin cofactors to react with a broad variety of epoxides, without the need of a catalysing enzyme, but still selecting for specific substitution patterns contributing to the correct steric and electronic environment. While the reaction unfortunately does not follow one single reaction path but leads to up to 6 different outcomes, the authors nicely illustrate the different pathways and do a great job on the structure elucidation of the large number of products they isolated. The products with expanded and contracted ring structures seem especially interesting to me, since those would not be easily achievable with established synthetic methods. Another very interesting finding is the peroxide **14**, and, although a stereochemical assignment of carbon number 3 would be nice to have, the existence of this intermediate and the prove of the further conversion to only a selection of the originally observed products arising from the FST C already gives it a very important role in the final mechanistic proposal. While the method as presented herein might not be of immediate use for the field of epoxide opening, a very important reaction in natural product total synthesis, it will surely inspire other groups to investigate the use of flavin cofactors in combination with (chiral) catalysts to direct it into one specific direction. Given this inspirational aspect and the extremely thorough and detailed work provided by the authors I recommend this publication to be accepted if some changes are made to present this on an even better and journal-appropriate level.*

Response 3. We would like to thank reviewer #3 for the constructive comments, which have helped us to improve the manuscript.

Comment 3.1. *While absolute amounts are given for the isolation of all compounds the yield is missing. This needs to be corrected.*

Response 3.1. The yields of all isolated compounds have been added to the Methods section (see **Preparation and isolation of 3, 7–11, 22–27 & 38–41**).

Comment 3.2. *Regarding the conversion of compound 42: did the authors consider different analytical techniques than TLC? While this is a very sensitive method, a spot on the same height, doesn't necessarily mean, that this is the same compound. Either a third lane with the putative product would be needed, or some other evidence in the form of a spectral analysis should be given.*

Response 3.2. We agree with reviewer #3 that the lack of a change in the TLC does not necessarily imply no reaction. Our efforts to develop an HPLC methodology to analyze the putative reaction of compound **42** with FAD and NADH were unsuccessful, likely on account of the long alkyl side chain in **42** resulting in strong retention on the reversed-phase C18 column. In an effort to provide additional evidence that **42** is nonreactive, we have included a characterization of the reaction by optical rotation and UV-vis absorbance showing no differences versus unreacted **42**. These data are now provided in **Supplementary Fig. 54** of the revised supporting information and reproduced below for convenience. We have also modified the main text to read more precisely as follows (see lines 237–238):

“However, incubation of **42** with FAD/NADH led to no changes in either its TLC retention profile or spectroscopic properties implying no reaction (see Supplementary Fig. 54).”

Supplementary Fig. 54 TLC profile for the reaction, UV spectra, and optical rotation of vitamin K1 2,3-epoxide (**42**) with FAD/NADH.

Comment 3.3. Presentation of NMR data (a-h):

Comment 3.3a. a lot of the NOESY data are not appropriately zoomed, that should be changed and the crosspeaks that were used for structure elucidation should be highlighted for an easier overview. Also, the NOESY spectrum of **10** is missing the structure of the associated compound.

Response 3.3a. All of the NOESY spectra have been zoomed as requested, we have also highlighted the crosspeaks, and the chemical shifts used for structure elucidation have been annotated. Likewise, the NOESY spectrum for **10** has been added. We agree that the presentation is significantly improved and easier to read following these changes.

Comment 3.3b. Some of the 1D spectra and the axial spectra on the 2D spectra are not zoomed to an appropriate level as well, for better interpretation by the reader that should be changed. Also, a baseline correction should be done for all spectra to make them look more

appealing.

Response 3.3b. We have zoomed the 1D and 2D spectra as requested and performed baseline corrections in order to improve the readability.

Comment 3.3c. *The NMR spectra for compound 9 look surprisingly bad, considering that 2.6 mg of this compound have been isolated, while other spectra for smaller amounts look much better. A clean spectrum of the compound would be nice.*

Response 3.3c. Compound 9 was reisolated and a set of better NMR spectra are provided.

Comment 3.3d. *For the deuteration experiments on compounds 7 and 8 it would be nice to show the ¹³C NMR spectra as additional proof for attachment of deuterium to carbon 3.*

Response 3.3d. We thank reviewer #3 for this suggestion. The ¹³C NMR spectra of deuterated 7 and 8 are now provided in the revised SI as **Supplementary Fig. 13–14**. Furthermore, a short note was added to the main text as follows (**see lines 139–141**),

“Moreover, the ¹³C NMR spectra of 7-²H and 8-²H both showed broadening and reduced intensity of the signals from C-3, and ²H–¹³C couplings at C-3 were also observed (Supplementary Figs. 13–14).”

Comment 3.3e. *For compound 14 additional spectra would be nice, if that is possible. If not, due to the instable nature a short comment would be great.*

Response 3.3e. Indeed, the instability of compound 14 made the collection of additional spectra difficult. We have added the following comment to point this out (see **lines 185–186**),

“; however, poor recovery of 14 hampered collection of sufficient spectra for a complete characterization of its absolute configuration at C-3.”

Comment 3.3f. *The ¹H-NMR of compound 25 is missing integrals*

Response 3.3f. Integrations have been added to the ¹H NMR spectrum of compound 25, and we thank the reviewer for pointing this out.

Comment 3.3g. *In some of the NOESY spectra the hydrogen at C1 is drawn with a relative*

instead of an absolute configuration, which should be adjusted.

Response 3.3g. The NOESY spectra of all compounds have been carefully checked and adjusted as requested.

Comment 3.3h. *The NOESY for compound 47 is missing. Although it is not very helpful for structure elucidation, it should still be presented as part of a complete dataset.*

Response 3.3h. The NOESY spectrum of compound 47 has been included.

Comment 3.4. *For all newly identified compounds the specific rotation should be measured and reported, as part of a full characterisation for a chiral compound.*

Response 3.4. Optical rotations were measured for 15 newly identified compounds; however, two new compounds (14 and 41) could not be isolated in sufficient quantity to measure the specific rotation. These data have been added to the section of “**Structural elucidation of reaction products**” in Supplementary text.

Comment 3.5. *For the conversion of compounds 45 and 46 it would be appropriate to briefly mention, that those are enantiomers and that it's hence not surprising, that only one is converted by a chiral catalyst, i.e., the enzyme RslO5.*

Response 3.5. This is a good point, and it is briefly mentioned as follows (lines 248–249),

“however, 46 but not 45 was accepted as a substrate by RlsO5 to yield the single product 47 (Supplementary Table 13 & Figs. 56–57), which indicates that RslO5 is specific for only one of the two enantiomers.”

Comment 3.6. *The mass spectra shown in the supporting information could be polished a little bit, m/z should be given in the appropriate form with “m” and “z” being in italics.*

Response 3.6. We have carefully checked all the mass spectra in the supporting information to ensure the appropriate formatting.

Comment 3.7. *Lastly, the manuscript gets very lengthy in parts, where methodologies for structure elucidation are described very precisely in the main text. Given the fact that this is supposed to be a communication, a more concise presentation would be appropriate, and some parts should be shortened to fit the manuscript into the 5000 word limit which is*

slightly exceeded.

Response 3.7. In an effort to reduce the total word count in the main text, we have moved many of the details regarding structural elucidation to the supporting information. The total word count is now reduced to approximately 5300.

Comment 3.8. *Apart from these, mostly cosmetic, corrections I found the manuscript very appealing to read (especially with the extraordinarily well organised and presented Figures!) and I think that it is of interest to a broad readership, from a biological, as well as from a chemical background.*

Response 3.8. We found these suggestions very helpful in improving the overall presentation of the manuscript and very much appreciate the reviewer's assistance and critical reading.

REVIEWERS' COMMENTS

Reviewer #1 (Remarks to the Author):

My concerns have been well addressed. Now it can be accepted.

Reviewer #3 (Remarks to the Author):

The authors did an outstanding job in their revision. All the requested points were immaculately and professionally addressed and additional/improved information was added to provide an even better manuscript. Especially the increase of information within the SI (60 additional pages!) shows the hard work that was put into providing a complete and extensive story.

From my point of view, this form of the manuscript is fitting to the high level of "Nature Communications" and I suggest this to be published without further changes. I am looking forward to see the final version of this magnificent piece of work!

Response to Reviewer Comments

Reviewer #1 (Remarks to the Author):

Comment:

My concerns have been well addressed. Now it can be accepted.

Response: We are glad that no further revisions are requested by Reviewer #1.

Reviewer #3 (Remarks to the Author):

Comment:

The authors did an outstanding job in their revision. All the requested points were immaculately and professionally addressed and additional/improved information was added to provide an even better manuscript. Especially the increase of information within the SI (60 additional pages!) shows the hard work that was put into providing a complete and extensive story.

From my point of view, this form of the manuscript is fitting to the high level of "Nature Communications" and I suggest this to be published without further changes. I am looking forward to see the final version of this magnificent piece of work!

Response: We appreciate reviewer #3 for the positive comments: no further revisions are requested.